# The prevalence and patterns of exposure to interpersonal violence among men and women in Estonia

Hedda Lippus[1,2]*, Kadri Soo[3], Made Laanpere[1,4,5], Kathryn M. Yount[2,6], Kai Part[1,4,5], Inge Ringmets[7], Mare Ainsaar[3], Helle Karro[1,4]

1 Department of Obstetrics and Gynaecology, Institute of Clinical Medicine, University of Tartu, Tartu, Estonia, 2 Hubert Department of Global Health, Rollins School of Public Health, Emory University, Atlanta, Georgia, United States of America, 3 Institute of Social Studies, University of Tartu, Tartu, Estonia, 4 Tartu University Hospital Women's Clinic, Tartu, Estonia, 5 Sexual Health Clinic of Tartu, Tartu, Estonia, 6 Department of Sociology, Emory University, Atlanta, Georgia, United States of America, 7 Institute of Family Medicine and Public Health, University of Tartu, Tartu, Estonia

* hedda.lippus@ut.ee

**Data Availability Statement:** All relevant data are within the paper and its Supporting Information files

## Abstract

### Background

To understand better the total burden of interpersonal violence on society, it is useful to assess the prevalence of interpersonal violence among both, men and women. Exposure to multiple forms of interpersonal violence, referred to as polyvictimization, has been associated with more severe health consequences than exposure to any form separately. The aims of this study were to assess the prevalence of emotional, physical and sexual interpersonal violence in childhood, adulthood and at both childhood and adulthood among men and women in Estonia, analyze the patterns of interpersonal violence and socio-demographic correlates of polyvictimization in adulthood by gender.

### Methods

The analysis was based on two population-based, cross-sectional, self-administered surveys carried out among men and women in Estonia in 2014. In both surveys, the NorVold Abuse questionnaire was used to measure exposure to interpersonal violence. Men and women aged 18–44 were included to the analysis.

### Results

Among men 66.6% and among women 54.2% had been exposed to at least one form of interpersonal violence during lifetime. Men had been more often exposed to isolated physical interpersonal violence, among women the distribution of different forms of interpersonal violence was more even and exposure to sexual violence was more common. The prevalence of polyvictimization in adulthood was two times higher among women compared to men and more socio-demographic correlates, were associated with it. Exposure to violence in childhood was associated with polyvictimization in adulthood across gender.

**Funding:** This study was funded by the European Regional Development Fund (https://ec.europa.eu/regional_policy/et/funding/erdf/) administered by the Estonian Research Council project TerVe (grant number 3.2.1002.11–0002), Institutional Research Funding IUT (https://www.etag.ee/rahastamine/uurimistoetused/institutsionaalne-uurimistoetus/) 34–16 and by the Government Office of Estonia (https://www.riigikantselei.ee/en/government-office). Funders did not play any role in the study design, data collection and analysis, decision to publish, or preparation of the manuscript.

**Competing interests:** The authors have declared that no competing interests exist.

## Conclusions

The prevalence of interpersonal violence in Estonia is high among men and women. The most prevalent forms and patterns of interpersonal violence, however, differ by gender, as do the socio-demographic correlates. Screening for interpersonal violence, in childhood and adulthood, and gender-specific interventions are needed, especially for high-risk groups identified in this study. Primary prevention of childhood violence should be a priority, as it was associated with higher risk for exposure to violence later in life across gender.

## Introduction

According to the World Health Organization, interpersonal violence refers to violence between individuals, and is subdivided into *family and intimate partner violence (IPV)*, and *community violence* [1]. Most of the research on interpersonal violence has focused on one of the types of interpersonal violence, like violence perpetrated by current of previous partner. However, many researchers have drawn attention to the fact, that leaving out other types of violence, where the perpetrator was somebody else than the intimate partner or that took place earlier in life, results in lower prevalence estimates and a limited understanding about the total burden of violence [2–5].

More recent research has demonstrated that different forms of violence rarely occur in isolation [2,6]. A large proportion of victims report being exposed to multiple forms of violence [2]. To better describe this phenomenon the term polyvictimization (PV) was coined. PV has been defined as experience of multiple victimizations of different kinds, not just multiple episodes of the same kind of victimization [4]. PV tends to be associated with more serious health consequences than exposure to a single violent event or recurrence of the same form of violence [2,7–9]. For example, a study carried out in Sweden showed that exposure to multiple forms of violence was more strongly associated with psychological ill-health than any single form of victimization among women and men [2]. The number of adverse childhood experiences, such as violence or neglect, has been shown to have a graded relationship to the presence of adult diseases [7]. Exposure to four or more adverse childhood experiences has been associated with leading causes of death in adults [7].

Population-based studies of the lifetime exposure to different forms of interpersonal violence among men and women are limited. Most research on the prevalence of interpersonal violence has focused on violence against women and on men's perpetration, in part because of the greater burden of certain forms of violence for women and the greater adverse effects of violence on women's mental and physical health [10,11]. Still, to understand the population burden of exposure to violence and the sub-populations who are at elevated risk, it is important to know the prevalence and socio-demographic factors associated with exposure to interpersonal violence among both men and women [2,7,12].

A higher prevalence of interpersonal violence is related to greater acceptance of violence in society, lower gender equality, and many other societal factors [1]. This study analyzes interpersonal violence in Estonian context, to understand the cultural and historical framework of this study, we provide here a brief overview about the recent history of Estonia, which is notable for various societal and political changes and is similar to other countries in the Eastern Europe. In 1940, Estonia was occupied by the Soviet Union. During that period, interpersonal violence was considered to be a personal issue not a public health concern meriting research

and interventions to prevent it. Due to that, there is no reliable data available on the prevalence of interpersonal violence from that period. Estonia regained independence in 1991, after which a period of transition to a democratic and free-market country followed. The restructuring period was accompanied by increase in unemployment, high mortality rates suicide and violent crime, similar tendencies have been observed also in other countries in transition [13–15]. For example, during the 1990s, murder rates in Estonia were among the highest in Europe [13], information concerning violence against women from this period is scarce, but some data suggest that the number of rapes increased [16]. Although substantial progress has been made since 1991, some historical legacies still are evident in today's society and the country has high scores in interpersonal violence scale [17]. For example, every fifth person in Estonia still considers family violence to be a private issue, and victim-blaming attitudes remain common [18]. Research looking at the prevalence and health consequences of violence has been emerging. A previous study demonstrated that in Estonia 17.2% of women had been exposed to physical and 4.1% to sexual IPV during the last year and it was an important contributor to sexual risk behaviour and adverse sexual health outcomes among women of reproductive age [19]. A study carried out by the European Agency for Fundamental rights showed that since age of fifteen every third woman in Estonia had been exposed to physical or sexual violence by partner/non-partner [20]. However, knowledge about both the prevalence and health consequences of interpersonal violence in Estonia and in Eastern Europe overall is limited in comparison with other European and North American countries. This study is the first in the Eastern European region to measure and compare the population-level prevalence of interpersonal violence among both, men and women and to look at the socio-demographic factors associated with exposure to polyvictimization in adulthood (PVA). The primary aim of this study was to describe the prevalence and co-occurrence of different forms of interpersonal violence in childhood and adulthood by gender. The second aim was to analyze the association between exposure to one form of violence, PVA and socio-demographic variables identified in the literature to be associated with a higher risk for exposure to interpersonal violence among men and women [1,21–24].

## Methods

### Procedures

This study was based on two cross-sectional surveys carried out in Estonia in 2014. Firstly, Estonian women´s health 2014: sexual and reproductive health, health behaviour, attitudes and use of healthcare services (hereafter Estonian Women's Health Survey, EWH) and secondly, Survey of Estonian men's attitude and behaviour: health, education, employment, migration and family formation (hereafter Estonian Men's Survey, EMS) [25,26]. In both surveys, several ethical considerations were followed. In the covering letter, all participants were informed the purpose of the study, how the findings will be used, instructions of filling, and contacts for obtaining additional information. Participation in the survey was voluntary and anonymous. The participants were free to withdraw their participation at any time or not to answer any particular questions. Moreover, the answers and the personal codes were stored in the separate databases to eliminate identification of certain individuals by their answers. The ethics committee approval is not required for carrying out anonymous questionnaire-based studies in Estonia, but Women's Health Survey was approved by the Research Ethics Committee of the University of Tartu, Estonia (226/T-7).

For the EWH and EMS, random samples of the female/male population, stratified by age groups were taken from the Estonian Population Registry. Power analysis was carried out for both surveys in order to determine optimal sample sizes. Based on the sample size calculations,

the initial sample size for EWS had to be 5233 women aged 16–44 and for EMS 4800 men aged 16–54. The sample size of EWS made up 2.1% of the total female population aged 16–44 years and sample size of EMS made up 1.5% of male population aged 16–54 living in Estonia in 2013. A total of 2440 women and 2119 men participated in these surveys, yielding response rates of 47.0% and 45.9%, respectively.

In the EWH, a self-completed survey method was used with the possibility to complete the questionnaire electronically or on paper [25]. All women in the sample were mailed a letter with a questionnaire and two pre-paid envelopes (one for returning the filled questionnaire and a second one including an individual code to let the researchers know that the respondent had posted the questionnaire and thus to allow the respondent's questionnaire to remain anonymous). The letter also included a link to an electronic questionnaire and personal code on the website. Of the respondents, 16.6% answered electronically and 83.4% on paper. There were no significant differences between the two responding methods regarding responses about exposure to violence or socio-demographic characteristics.

In the EMS, eligible respondents were mailed a letter, including a link to electronic questionnaire and a personal code to access the online questionnaire [26]. On request, a paper questionnaire with a prepaid envelope to return it, and a card with a personal code which had to be posted separately, was available. The researchers visited men who had not responded by either paper or electronic questionnaire after three reminders. They gave out questionnaires and later collected filled in ones in sealed envelopes. Most participants responded electronically (93%), 2.2% returned the questionnaire by post, and 4.8% by the researcher. In the WHS, LimeSurvey program was used, but in the EMS, SurveyMonkey platform was used. Ethical guidelines for research on violence were followed in both surveys [27]. More detailed descriptions concerning the methodology of the surveys can be found from the survey reports [25,26].

## Sample

Men (n = 611) and women (n = 749) either over 44 or under 18 years old, or who did not respond to the question about age were excluded from the analysis to allow comparisons across gender. Respondents who had not answered the questions about native language (23 women) or violence exposure (264 men and 78 women) were also excluded. Thus, the final sample for analysis included 1244 men and 1590 women. The data for both surveys were weighted by age group and native language (Estonian/Russian or other) using census data [28]. Weighting was carried out to compensate for oversampling of younger women and lower response rates of Russian-speaking men and women. Information in this study concerning the respondents' gender comes from the population registry, where it is recorded either as male or female.

## Measures

**Violence assessment.**   In both surveys, the NorVold Abuse Questionnaire (NorAQ) was used to assess exposure to interpersonal violence in childhood and adulthood [29,30]. NorAQ has been validated in male and female samples and has shown adequate test-retest reliability, sensitivity (68–96%) and specificity (72–99%), except for the question about mild physical violence, which was excluded from the present study [2,29,30]. Questions in NorAQ are behaviourally specific–this means that the respondents could choose from pre-defined answer alternatives describing violent behavior. There were four answer alternatives which were same for all questions: 1) no; 2) as a child (before the age of 18); 3) yes, as an adult (when being 18 years old or older) 4) yes, as a child and as an adult (before and after the age of 18). In Table 1, questions and answer choices in NorAQ are presented [29].

**Table 1. Questions about exposure to violence in NorVold Abuse questionnaire.**

| **Emotional violence** | |
|---|---|
| Mild | Have you experienced anybody systematically and for any longer period trying to repress, degrade or humiliate you? |
| Moderate | Have you experienced anybody systematically and by threat or force trying to limit your contacts with others or totally control what you may and may not do? |
| Severe | Have you experienced living in fear because somebody systematically and for a longer period has threatened you or somebody close to you? |
| **Physical violence** | |
| Moderate | Have you experienced anybody hitting you with his/her fist(s) or with a hard object, kicking you, pushing you violently, giving you a beating, thrashing you or doing anything similar to you? |
| Severe | Have you experienced anybody threatening your life by, for instance, trying to strangle you, showing a weapon or knife or by any other similar act? |
| **Sexual violence** | |
| Mild, no genital contact | Has anybody against your will touched parts of your body other than the genitals in a 'sexual way' or forced you to touch other parts of his or her body in a 'sexual way'? |
| Mild, emotional/ sexual humiliation | Have you in any other way been sexually humiliated; e.g. by being forced to watch a porno movie or similar against your will, forced to participate in a porno movie or similar, forced to show your body naked or forced to watch when somebody else showed his/her body naked? |
| Moderate, genital contact | Has anybody against your will touched your genitals, used your body to satisfy him/herself sexually or forced you to touch anybody else's genitals? |
| Severe, penetration | Has anybody against your will put his penis into your vagina*, mouth or rectum or tried any of this; put in or tried to put an object or other part of the body into your vagina, mouth or rectum? |
| Answer alternatives (the same for all questions) | |
| 1 = No | |
| 2 = Yes, as a child (<18 years) | |
| 3 = Yes, as an adult (≥18 years) | |
| 4 = Yes, as a child and as an adult | |

*The word "vagina" omitted from men's questionnaire

To analyze the patterns of exposure to interpersonal violence, we created two aggregate measures. In order to assess the exposure to different forms of interpersonal violence and their combinations in childhood, adulthood and both in childhood and adulthood, a new variable was created, where respondents were grouped into eight categories as follows: 1. no violence; 2. emotional; 3. physical; 4. sexual; 5. emotional and physical; 6. emotional and sexual; 7. physical and sexual; 8. emotional, physical and sexual. Then, polyvictimization in adulthood (PVA) was created by dividing the participants into three groups based on the number of forms of violence they had been exposed to as adults. Both, respondents who had been exposed to violence only in adulthood and those who had been exposed to violence both in childhood and adulthood were included in this variable. The groups are: 1. no exposure in adulthood 2. exposure to one form of violence 3. exposure to two or three forms of violence.

**Independent variables in the multinomial logistic regression model.** Ten variables were included in the analysis, separately for men and women: native language (Estonian; Russian/ other); level of education (primary or less; secondary/vocational secondary; higher; missing); education of the mother and of the father (unknown; primary or less/secondary; higher; missing); marital status (married; cohabiting; single; other; missing); having one or more children (no; yes); estimation of financial situation (very good/good; neither good nor bad; bad/very bad; missing); sexual orientation (exclusively or predominantly heterosexual, bi- or

homosexual; missing), exposure to violence in childhood (no; one form of violence; polyvicti-mization) and age in years (18–44 years old).

## Analysis

Descriptive statistics were used to describe the prevalence of the three different forms of violence, and the socio-demographic characteristics of the respondents. Differences in the prevalence of emotional, physical and sexual violence and their combinations across gender was analyzed using a chi-square test with a significance level of p<0.01. Multinomial logistic regression analyses (adjusted for covariates) were estimated for women and men separately to examine the associations of socio-demographic characteristics (independent variables) with exposure to 1) one form of violence and 2) PVA (dependent variables). All respondents who did not answer some of the questions in this study were excluded from the models, the responses of 1239 men and 1532 women were included to the multinomial logistic regression analysis.

## Results

### Socio-demographic characteristics of respondents

As shown in Table 2, having higher education (24.7% *vs*. 42.6%) being officially married (27.2% *vs*. 31.8%), and having at least one or more children (43.6% *vs*. 59.4%) was less common among men compared to women. Men estimated their financial situation more often to be good or very good than women (16.7% *vs*. 10.0%). Nearly 98% of both men and women reported to be exclusively or predominantly heterosexual.

### Prevalence of interpersonal violence

Two thirds (n = 829, 66.6%) of men and over half (n = 862, 54.2%) of women had been exposed to some form of interpersonal violence in their lifetime (Table 3). The patterns of exposure to interpersonal violence among men and women were statistically significantly different across all life periods. Exposure to physical violence only was the most common among men in all groups. In childhood, exposure to emotional violence only was the most common among women (12.3%), however it remained somewhat lower than among men (14.5%). Exposure to sexual violence among women was more common than among men in all groups. In childhood 5.8% and both in childhood and adulthood 0.9% of women had been exposed to sexual violence, which means that in total almost seven percent of women had been exposed to sexual violence in childhood. Apart from the combination of emotional and physical violence, exposure to multiple forms of interpersonal violence among men remained under one percent in all groups. Among women the co-occurrence of different forms was more common, and exposure was more evenly distributed between different forms.

### Multinomial logistic regression results

Among men, 28.9% (n = 358) of the respondents were exposed to one form of interpersonal violence and 5.8% (n = 72) to PVA in adulthood or both in childhood and adulthood, among women 17.9% (n = 274) and 12.4% (n = 190), accordingly. In multinomial logistic regression analysis, exposure to violence in childhood and older age were associated with exposure to one form of interpersonal violence across gender, among women it was additionally associated with cohabiting or being divorced/widowed (Table 4). Among men, unknown education of mother, neutral or poor financial situation, exposure to violence in childhood and older age

**Table 2. Weighted socio-demographic characteristics by gender, 18–44-year-old respondents in Estonia, %.**

| Socio-demographic characteristics | Men | Women |
|---|---|---|
| | **n = 1244** | **n = 1590** |
| **Age** | | |
| 18–24 | 24.2 | 25.1 |
| 25–34 | 38.5 | 37.9 |
| 35–44 | 37.3 | 37.0 |
| **Native language** | | |
| Estonian | 74.9 | 70.5 |
| Russian or other | 25.1 | 29.5 |
| **Education** | | |
| Primary education or less | 13.7 | 15.6 |
| Secondary or vocational secondary education | 61.5 | 41.3 |
| Higher education | 24.7 | 42.6 |
| Missing | 0.1 | 0.5 |
| **Education of mother** | | |
| Unknown | 4.8 | 2.0 |
| Primary education or less/secondary education | 62.2 | 66.6 |
| Higher education | 33.0 | 31.4 |
| Missing | 0.0 | 0.4 |
| **Education of father** | | |
| Unknown | 11.7 | 9.5 |
| Primary education or less/ Secondary education | 60.1 | 65.5 |
| Higher education | 28.2 | 24.6 |
| Missing | 0.0 | 0.4 |
| **Marital status** | | |
| Married | 27.2 | 31.8 |
| Cohabiting | 36.4 | 39.0 |
| Single | 32.0 | 23.9 |
| Other | 4.4 | 5.0 |
| Missing | 0.0 | 0.3 |
| **Having one or more children** | | |
| No | 56.4 | 40.6 |
| Yes | 43.6 | 59.4 |
| **Estimation on financial situation** | | |
| Very good or good | 16.7 | 10.0 |
| Neither good nor bad | 49.5 | 48.4 |
| Very bad or bad | 33.8 | 40.7 |
| Missing | | 0.9 |
| **Sexual orientation** | | |
| Exclusively or predominantly heterosexual | 97.7 | 97.6 |
| Bi- or homosexual | 2.0 | 1.7 |
| Missing | 0.3 | 0.7 |

were associated with PVA. Exposure to PVA among women was positively associated with all covariates, except for education of father and sexual orientation. Among both men and women polyvictimization in childhood had graded relationship with PVA.

**Table 3. Weighted prevalence of the co-occurrence of different forms of violence during childhood, adulthood, both childhood and adulthood and lifetime exposure by gender, 18–44-year-old men (n = 1244) and women (n = 1590) in Estonia, %.**

| Exposure to interpersonal violence | Only in childhood* | | Only in adulthood* | | Both in childhood and adulthood* | | Lifetime exposure* | |
|---|---|---|---|---|---|---|---|---|
| | Men % | Women % | Men % | Women % | Men % | Women % | Men % | Women % |
| No exposure to any form | 52.7 | 61.6 | 78.0 | 78.1 | 82.5 | 90.8 | 33.4 | 45.8 |
| Only emotional | 14.5 | 12.3 | 2.8 | 4.5 | 3.5 | 4.1 | 8.8 | 10.8 |
| Only physical | 17.8 | 7.7 | 15.9 | 5.7 | 11.4 | 1.9 | 30.0 | 10.1 |
| Only sexual | 0.6 | 5.8 | 1.0 | 4.2 | 0.0 | 0.9 | 0.2 | 6.2 |
| Emotional and physical | 12.4 | 5.3 | 1.8 | 3.6 | 2.5 | 1.1 | 23.6 | 10.4 |
| Emotional and sexual | 0.8 | 2.7 | 0.0 | 1.4 | 0.0 | 0.4 | 0.4 | 3.8 |
| Physical and sexual | 0.5 | 2.0 | 0.2 | 1.1 | 0.0 | 0.2 | 1.0 | 3.8 |
| Exposure to all three forms (emotional, physical and sexual) | 0.7 | 2.7 | 0.2 | 1.4 | 0.1 | 0.6 | 2.8 | 9.1 |

*Statistically significant differences between men and women according to chi-square analysis, p<0.01

## Discussion

The results of this study show that more than half of men and women have been exposed to at least one form of interpersonal violence, however the patterns of exposure are significantly different. Among women the distribution of different forms of violence is more even and exposure to PVA is higher than among men. In addition to that, the results suggest that among women PVA is associated with more socio-demographic characteristics.

Findings from prior studies carried out in other developed countries, which have looked at gender differences in the context of IPV or violence in childhood, have showed different patterns of exposure to violence across gender [31–33]. For example, some studies have suggested that men are exposed more often than women to physical violence in childhood and youth [33,34] and women are exposed more often than men to sexual IPV and sexual youth violence [12,31,33], which corroborates the findings of this study.

This is the first study in Estonia and Eastern European region demonstrating the vast differences in the patterns of exposure to interpersonal violence among men and women across childhood and adulthood. The results of this study show that among men, physical interpersonal violence often occurs in isolation, which is in line with previous findings [33]. The exposure to violence during adolescence among men has been related to higher rates of conventional crime, whereas among women rates of relational violence are higher [35]. Empirically, boys have received more harsh verbal and physical punishments from their parents and/ or caregivers [34], which can partly explain this finding. Among boys physical violence at school has been shown to be more common [36]. Among women, exposure to sexual interpersonal violence only or in combination with emotional and/or physical violence was the highest in childhood and significantly more common than among men. Childhood and youth have been shown to be periods with higher risk for sexual violence exposure, similarly with the results of this study, girls have been found to have even higher risk than boys [1,33,37].

According to our results, every sixth man has been exposed to physical interpersonal violence in adulthood, while the rates of other forms of violence remain significantly lower compared to women. There is evidence showing that men are more often exposed to community violence, such as violent crimes [38]. Among women, exposure to different forms of interpersonal violence is distributed more evenly.

Exposure to all three forms of interpersonal violence in childhood was three times and in adulthood seven times more common among women than men. Exposure to

**Table 4. Adjusted odds ratios (AOR)\* with 95% confidence intervals for being exposed to one form of violence and polyvictimization in adulthood, 18–44-year-old respondents in Estonia.**

| Background characteristics | Men n = 1239 | | Women n = 1532 | |
|---|---|---|---|---|
| | One form of violence in adulthood** n = 360 (28.9%) | Polyvictimization in adulthood** n = 72 (5.8%) | One form of violence in adulthood** n = 285 (17.9%) | Polyvictimization in adulthood** n = 197 (12.4%) |
| **Native language** | | | | |
| Estonian | 1.00 | 1.00 | 1.00 | 1.00 |
| Russian/other | 0.97 (0.72–1.32) | 0.86 (0.48–1.56) | 1.08 (0.82–1.42) | **1.45 (1.06–1.99)** |
| **Respondents education** | | | | |
| Higher | 1.00 | 1.00 | 1.00 | 1.00 |
| Basic | 1.42 (0.90–2.24) | 1.22 (0.48–3.09) | 1.36 (0.90–2.05) | **2.56 (1.57–4.18)** |
| Secondary | 1.12 (0.82–1.54) | 1.15 (0.60–2.21) | 1.12 (0.86–1.47) | **2.01 (1.42–2.83)** |
| **Education of mother** | | | | |
| Higher | 1.00 | 1.00 | 1.00 | 1.00 |
| Basic/ secondary | 0.96 (0.70–1.30) | 1.62 (0.83–3.17) | 0.87 (0.65–1.16) | **1.58 (1.07–2.36)** |
| Unknown | 1.53 (0.79–2.98) | **3.84 (1.17–12.60)** | 1.63 (0.65–4.03) | **3.39 (1.33–8.65)** |
| **Education of father** | | | | |
| Higher | 1.00 | 1.00 | 1.00 | 1.00 |
| Basic/ secondary | 1.22 (0.88–1.68) | 0.85 (0.44–1.63) | 1.01 (0.74–1.39) | 0.88 (0.59–1.33) |
| Unknown | 1.40 (0.87–2.27) | 0.86 (0.34–2.20) | 0.78 (0.46–1.30) | 0.72 (0.40–1.32) |
| **Marital status** | | | | |
| Married | 1.00 | 1.00 | 1.00 | 1.00 |
| Single | 1.29 (0.85–1.96) | 1.87 (0.82–4.26) | 1.29 (0.86–1.94) | **1.74 (1.07–2.82)** |
| Cohabiting | 1.35 (0.95–1.92) | 1.28 (0.63–2.58) | **1.65 (1.20–2.26)** | **1.97 (1.36–2.86)** |
| Divorced/widowed/other | 1.59 (0.84–3.02) | 1.86 (0.57–6.09) | **2.21 (1.30–3.77)** | **2.28 (1.25–4.16)** |
| **Biological children** | | | | |
| No | 1.00 | 1.00 | 1.00 | 1.00 |
| Yes | 0.74 (0.53–1.04) | 1.21 (0.61–2.42) | 0.83 (0.59–1.17) | **1.75 (1.13–2.71)** |
| **Estimation of financial situation** | | | | |
| Good/very good | 1.00 | 1.00 | 1.00 | 1.00 |
| Neither good nor bad | 1.24 (0.93–1.65) | **2.24 (1.09–4.58)** | 1.17 (0.90–1.51) | 1.19 (0.84–1.66) |
| Bad/very bad | 1.27 (0.86–1.88) | **4.16 (1.18–9.19)** | 1.43 (0.92–2.20) | **3.26 (2.09–5.09)** |
| **Sexual orientation** | | | | |
| Exclusively or predominantly heterosexual | 1.00 | 1.00 | 1.00 | 1.00 |
| Bi- or homosexual | 0.33 (0.09–1.20) | 2.51 (0.72–8.74) | 0.63 (0.22–1.74) | 1.76 (0.69–4.46) |
| **Exposure to violence in childhood** | | | | |
| No | 1.00 | 1.00 | 1.00 | 1.00 |
| One form of violence in childhood | **2.18 (1.63–2.90)** | **3.77 (1.76–8.08)** | **1.86 (1.43–2.50)** | **2.23 (1.59–3.13)** |
| Polyvictimization in childhood | **2.26 (1.61–3.16)** | **11.50 (5.60–23.58)** | **3.09 (2.26–4.23)** | **3.38 (2.33–4.90)** |
| **Age** | | | | |
| 18–44 | **1.05 (1.03–1.07)** | **1.05 (1.01–1.09)** | **1.05 (1.03–1.08)** | **1.05 (1.03–1.08)** |

\* Adjusted for covariates

\*\* Reference category is "No exposure to violence during adulthood"

polyvictimization has been characterized as living in a constantly unsafe environment, where there is no place where one could feel safe. In childhood, it has to do with dysfunctional households, poor parent/caregiver relationship and in adulthood with IPV [21,33]. In both of these situations the victim lives in a constant state of fear, which is believed to lead to the significantly higher levels of negative health outcomes, than exposure to any form of violence in isolation.

In addition to demonstrating the high prevalence of interpersonal violence among men and women in Estonia, the contribution of this study is identifying socio-demographic factors associated with higher risk for experiencing violence by gender. Some of the factors associated with PVA among men and women were overlapping, such as exposure to violence in childhood and older age, it should be noted that across gender exposure to violence in childhood had a graded relationship with PVA. Both of these factors have been shown in prior research to be strongly associated with exposure to IPV among women [39]. Although younger age *per se* is a well-known risk factor for experiencing violence, simply the longer time period over which one can be exposed to violence, results in higher lifetime prevalence rates in older age groups. Exposure to violence in childhood has been associated previously with both higher rates of revictimization and higher risk of violence perpetration [10,39]. There are several pathways which are hypothesized to cause higher rates of revictimization, it has been shown that children exposed to PV have a reduced capacity for affect regulation in adulthood, dysfunctional behavioural patterns, lower self-esteem and higher levels of psychological distress [40–42]. Children exposed to abuse and neglect are at increased risk for substance abuse and criminal behavior [24,40], which can lead both to being victimized of perpetration of violence. The instrument used in this study to measure exposure to violence did not contain questions about the perpetrator, but the strong association between exposure to violence in childhood and PVA combined with the high prevalence of violence in childhood suggests that this topic merits further investigation. Lower education of mother and belonging to lower socio-economic class were associated with higher risk for PVA both among men and women. In Estonia until quite recently, gender stereotypic distribution of household chores has been dominating [43], women have been responsible for taking care of the children, which can explain why the education of mother is associated with PVA, while unknown or low education of father is not. Unknown or low education of mother can mean growing up in lower-resource and unsafe setting and less knowledgeable child rearing practices. Parental education has also been shown to predict the child's future socioeconomic status, which in the current study was also associated with PVA [44]. Non-Estonian ethnicity, lower education, non-married status, and having one or more children were factors that were associated with PVA only among women. The large number of different background characteristics associated with exposure PVA among women shows that socio-demographic background plays much bigger role in exposure to violence, especially to PVA, among women than men. This knowledge can be used to design more evidence-based prevention and intervention strategies to prevent negative health outcomes shown in previous research to be associated with PVA.

This study has several limitations which should be addressed. Firstly, the questionnaire did not ask about the perpetrator of violence, neither the duration of exposure to violence. Information about the perpetrator could have helped to give more thorough explanations regarding the differences of the violence patterns between men and women. From previous research it is known that women are more often exposed to systematic forms of violence, which could explain the higher levels of polyvictimization among them [45]. Future research is needed in Estonia to evaluate the perpetrator of violence, the context where violence took place–was it one-off event or repeated violence—and the impact it has on the victim. Secondly, violence may still be associated with social stigma, especially some forms like sexual violence among

men, which can cause underreporting. Thirdly, many forms of violence, for example, economic violence, witnessing violent events, are not included in the NorAQ. The response rates of both surveys, 47% in EWH and 44.5% in EMS were in the range expected and similar response-rate estimations were used when calculating the sample size. In addition to that, the final samples of both EWH and EMS were representative and had no selectivity bias. Previous population-based studies focusing on violence have yielded similar response rates [20,32]. The prevalence of interpersonal violence in EWH survey is comparable with the results of the previous studies carried out in Estonia [20,46]. However, there is a possibility that the most highly victimized men and women were not able or chose not to participate in the surveys. Based on that, it is possible, that the prevalence of PVA could be even higher than shown in this paper.

One of the major strengths of this study is that the data came from two population-based cross-sectional studies and the use of the identical validated questionnaire. The questions concerning violence were filled in rather well (over 95% of the respondents answered to the questions concerning exposure to violence), taking into account the sensitivity of these questions.

These results offer valuable knowledge to public health and education specialists, healthcare providers and policy-makers, who can use this information to design interventions to prevent violence or develop violence screening programs. In addition to Estonia, these results are particularly useful in the Eastern European region, where until now research demonstrating the prevalence of interpersonal violence has been scarce. It is of utmost importance to pay more attention to prevent violence in childhood, and identify children who have been exposed to violence, as violence in childhood is strongly associated with increased risk of exposure to violence later in life across gender. For example, currently comprehensive violence prevention programs at schools and kindergartens are being developed in Estonia. The results of this study demonstrate significant differences between men and women in the violence exposure patterns. It should be acknowledged that women are twice as likely to be polyvictimized in adulthood than men and have more socio-demographic factors associated with it. Given the high number of people exposed to PVA and the magnitude of negative health consequences associated with it, the development of effective interventions in order to reduce its negative consequences should be prioritized. Our study provides essential information in the context of Estonia, but also contributes to the international knowledge about exposure to violence and the different patterns of violence among men and women. Violence is worryingly common in Estonia and can no longer be seen as a trivial problem.

## Supporting information

**S1 Data.**
(XLSX)

## Author Contributions

**Conceptualization:** Hedda Lippus, Kadri Soo, Made Laanpere, Mare Ainsaar, Helle Karro.

**Data curation:** Hedda Lippus, Kadri Soo, Inge Ringmets.

**Formal analysis:** Hedda Lippus, Kadri Soo, Inge Ringmets.

**Funding acquisition:** Made Laanpere, Mare Ainsaar, Helle Karro.

**Methodology:** Kadri Soo, Inge Ringmets.

**Supervision:** Made Laanpere, Kathryn M. Yount, Helle Karro.

**Writing – original draft:** Hedda Lippus.

**Writing – review & editing:** Kadri Soo, Made Laanpere, Kathryn M. Yount, Kai Part, Mare Ainsaar, Helle Karro.

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
