## [Decision Letter · Decision Letter 0]

5 Feb 2020

PONE-D-19-24414

The prevalence and patterns of exposure to interpersonal violence among men and women in Estonia

PLOS ONE

Dear Dr Lippus,

Thank you for submitting your manuscript to PLOS ONE. After careful consideration, we feel that it has merit but does not fully meet PLOS ONE’s publication criteria as it currently stands. Therefore, we invite you to submit a revised version of the manuscript that addresses the points raised during the review process. Please pay special note to the comments of reviewer 3, presented in the attachement.

We would appreciate receiving your revised manuscript by Mar 21 2020 11:59PM. To enhance the reproducibility of your results, we recommend that if applicable you deposit your laboratory protocols in protocols.io, where a protocol can be assigned its own identifier (DOI) such that it can be cited independently in the future. For instructions see: http://journals.plos.org/plosone/s/submission-guidelines#loc-laboratory-protocols

We look forward to receiving your revised manuscript.

Kind regards,

Astrid M. Kamperman

Academic Editor

PLOS ONE

Journal Requirements:

2. Please include additional information regarding the justification for sample size in this study. How was this sample size chosen - e.g. following a power calculation.

Reviewers' comments:

Reviewer's Responses to Questions

**Comments to the Author**

1. Is the manuscript technically sound, and do the data support the conclusions?

Reviewer #1: Yes

Reviewer #2: Yes

Reviewer #3: Partly

2. Has the statistical analysis been performed appropriately and rigorously? 

Reviewer #1: No

Reviewer #2: Yes

Reviewer #3: No

3. Have the authors made all data underlying the findings in their manuscript fully available?

Reviewer #1: Yes

Reviewer #2: Yes

Reviewer #3: No

4. Is the manuscript presented in an intelligible fashion and written in standard English?

Reviewer #1: No

Reviewer #2: Yes

Reviewer #3: Yes

5. Review Comments to the Author

Reviewer #1: This study explored interpersonal violence issue in an understudied population. Overall, the story is not cohesive. It took multiple approaches to investigate interpersonal violence, such as different forms of interpersonal violence, polyvictimization in adulthood, and exposure to violence during last year, while the multinomial logistic regression only tested polyvictimization in adulthood. Below, I offer some suggestions in the hope of strengthening this paper’s contribution.

Introduction

1 What is the rationale to include "different forms of interpersonal violence", "polyvictimization in adulthood", and "exposure to violence during last year" in one manuscript?

2 In the introduction, I'd like to see literature review on how childhood interpersonal violence experience will affect adulthood interpersonal violence.

3 A more detailed literature review on how to define polyvictimization is needed.

Methods

4 why didn't use regression with interaction term to test gender difference?

Discussion

5 The key finding is those who experienced childhood violence are more likely to experience polyvictimization in adulthood, which is innovative. This result should be better interpreted in the discussion.

6 P19, Para 2, "...but the strong association between PVA and exposure to violence in childhood..." it should be "...association between exposure to violence in childhood and PVA"

Table

7 There are two Table 1 in the manuscript

8 Table 3 didn't talk to Table 4. It is better to show the frequency of no interpersonal violence, one form of interpersonal violence in adulthood, and polyvictimization in adulthood between men and women in Table 3. Also, show exposure to violence in childhood in Table 3.

9 There are many parts in the manuscript using "violence", which is not rigorous. Violence is a very broad concept. It is better to be replaced by "interpersonal violence".

Reviewer #2: There are many typographical and grammatical errors throughout the manuscript.

1. Page 11, 3rd paragraph – The authors should ensure there description of results matches the name of variable levels in the table, e.g. “one child” should be “one biological child” and “good” should be “good or very good”.

2. Page 12, 1st paragraph – The authors fail to include “than women” and “than men” when making comparison throughout the manuscript.

3. Page 13, 1st paragraph – The authors should include the number of men and women who had been exposed to violence along with the percentage.

4. Page 14, 2nd paragraph – The authors should insert “during the last year” when describing table 3 results.

5. Page 15, 1st paragraph – The authors should indicate what questionnaire was used to examine violence in childhood.

6. Page 15, 2nd paragraph – The authors should delete their description of significant findings for age in table 4 since they are borderline.

7. Table 4 – The authors should include the numbers along with the percentages of men and women who were affected by violence. The authors should include a footnote indicating the variables used for adjustment.

Reviewer #3: 1. The authors could expound upon why they found heterogeneity in violence between men and women. As is, it is unclear how this study contributes to the literature.

2. The attached review includes recommendations on more rigorous analyses.

3. Data do not appear to be available

4. No additional comments

6. PLOS authors have the option to publish the peer review history of their article (what does this mean?). If published, this will include your full peer review and any attached files.

Reviewer #1: No

Reviewer #2: No

Reviewer #3: No

---

## [Author Response · Author response to Decision Letter 0]

24 Apr 2020

Journal Requirements:

Authors appreciate this comment and have followed the PLOS ONE style templates to meet the style requirements.

2. Please include additional information regarding the justification for sample size in this study. How was this sample size chosen - e.g. following a power calculation.

Authors are grateful for pointing this out. We have added the following sentences to the methods section regarding the justification of the sample sizes: 

“Power analysis was carried out for both surveys in order to determine optimal sample sizes. Based on the sample size calculations, the initial sample size for EWS had to be 5233 women aged 16-44 and for EMS 4800 men aged 16–54.” 

In addition to that, as both of the surveys used in this paper have published survey reports with detailed description of the methodology and both are available online, we added also the following sentence:

“More detailed descriptions concerning the methodology of the surveys can be found from the survey reports”.

Authors appreciate this comment. The phrase “data not shown” was removed form the manuscript. All relevant data is presented in the manuscript. 

Reviewer #1:

This study explored interpersonal violence issue in an understudied population. Overall, the story is not cohesive. It took multiple approaches to investigate interpersonal violence, such as different forms of interpersonal violence, polyvictimization in adulthood, and exposure to violence during last year, while the multinomial logistic regression only tested polyvictimization in adulthood. Below, I offer some suggestions in the hope of strengthening this paper’s contribution.

Authors are very grateful for these comments, which help us to improve the quality of the manuscript. We have taken them into account and carried out changes accordingly.

Introduction

1 What is the rationale to include "different forms of interpersonal violence", "polyvictimization in adulthood", and "exposure to violence during last year" in one manuscript?

Authors appreciate this comment. To make the paper more cohesive we have decided to leave out "exposure to violence during last year" from the manuscript.

The decision to show different forms of violence separately is associated with the definition of polyvictimization: “Polyvictimization refers to the experience of multiple victimizations of different kinds (Turner, Shattuck, Finkelhor, & Hamby, 2017)”. By showing the prevalence of different forms of interpersonal violence, the authors want to give a more in-depth overview to the readers about the formation of the “polyvictimization in adulthood” variable. However, we have changed Table 3 so that it shows the prevalence and combinations of different forms of violence in childhood, adulthood and childhood and adulthood and as previously mentioned left out “exposure to violence during last year”. By carrying out this change, we wanted to make data presented in Table 3 more connected to the following Table 4. 

Authors decided to use polyvictimization in the logistic regression model (not different forms of violence separately) because we wanted to focus on determining the risk factors associated with the most serious victimization profile, as in previous studies it has been shown to have the most detrimental health effects.

2 In the introduction, I'd like to see literature review on how childhood interpersonal violence experience will affect adulthood interpersonal violence.

Authors are grateful for this comment. We have taken this into account and carried out changes, this topic is more thoroughly addressed in the discussion. 

3 A more detailed literature review on how to define polyvictimization is needed.

We appreciate this comment. The definition of polyvictimization was added as follows:

“PV has been defined as experience of multiple victimizations of different kinds, not just multiple episodes of the same kind of victimization (Turner et al., 2017).”

Methods

4 why didn't use regression with interaction term to test gender difference?

The authors appreciate this comment. Authors chose not to use regression with interaction based on the assumption that factors associated with exposure to PVA are different among men and women and the aim of this paper was to focus on showing these differences. In addition to that, we wanted to keep the models more easily interpretable for the readers. 

Discussion

5 The key finding is those who experienced childhood violence are more likely to experience polyvictimization in adulthood, which is innovative. This result should be better interpreted in the discussion.

Authors appreciate this comment, based on that we have added a more thorough discussion as follows:

“There are several pathways which are hypothesized to cause higher rates of revictimization, it has been shown that children exposed to PV have a reduced capacity for affect regulation in adulthood, dysfunctional behavioural patterns, lower self-esteem and higher levels of psychological distress (Briere, Hodges, & Godbout, 2010; Chan, Brownridge, Yan, Fong, & Tiwari, 2011; Dugal, Godbout, Bélanger, Hébert, & Goulet, 2018). Children exposed to abuse and neglect are at increased risk for substance abuse and criminal behavior (Briere et al., 2010; Dovran et al., 2019), which can lead both to being victimized of perpetration of violence.”

6 P19, Para 2, "...but the strong association between PVA and exposure to violence in childhood..." it should be "...association between exposure to violence in childhood and PVA"

The authors agree with this comment and have changed the order as suggested by the reviewer.

Table

7 There are two Table 1 in the manuscript

The authors are grateful for pointing out the incorrect headings of the tables and have corrected the mistake. 

8 Table 3 didn't talk to Table 4. It is better to show the frequency of no interpersonal violence, one form of interpersonal violence in adulthood, and polyvictimization in adulthood between men and women in Table 3. Also, show exposure to violence in childhood in Table 3.

The authors appreciate this comment and have carried out changes to the Table 3. We now present in Table 3 no exposure to any form, exposure to only emotional, only physical, only sexual interpersonal violence and their combinations in childhood, adulthood and in childhood and adulthood. 

9 There are many parts in the manuscript using "violence", which is not rigorous. Violence is a very broad concept. It is better to be replaced by "interpersonal violence".

Authors highly appreciate this comment and have carried out changes in the manuscript.

Reviewer #2:

There are many typographical and grammatical errors throughout the manuscript.

The authors are grateful to the reviewer for this comment and have checked the manuscript rigorously to correct all typographical and grammatical errors.

1. Page 11, 3rd paragraph – The authors should ensure there description of results matches the name of variable levels in the table, e.g. “one child” should be “one biological child” and “good” should be “good or very good”.

The authors appreciate pointing out this inconsistency in the manuscript and have changed this in the manuscript. 

2. Page 12, 1st paragraph – The authors fail to include “than women” and “than men” when making comparison throughout the manuscript.

We appreciate this comment and have carried out changes in the manuscript accordingly. For example:

“As shown in Table 2, having higher education (24.7% vs. 42.6%) being officially married (27.2% vs. 31.8%), and having at least one biological child (43.6% vs. 59.4%) was less common among men compared to women. Men estimated their financial situation more often to be good or very good than women (16.7% vs. 10.0%)”

3. Page 13, 1st paragraph – The authors should include the number of men and women who had been exposed to violence along with the percentage.

The authors appreciate this comment and have added the numbers of men and women as follows: 

“Two thirds (n=829, 66.6%) of men and over half (n=862, 54.2%) of women had been exposed to some form of interpersonal violence in their lifetime.”

4. Page 14, 2nd paragraph – The authors should insert “during the last year” when describing table 3 results.

We appreciate this comment. Table 3 has been changed and exposure to violence during last year has been omitted form the manuscript. 

5. Page 15, 1st paragraph – The authors should indicate what questionnaire was used to examine violence in childhood.

Authors appreciate this comment. The NorVold Abuse Questionnaire was used to measure exposure to violence in childhood and adulthood. We have added information regarding that to the methods section as follows:

“In both surveys, the NorVold Abuse Questionnaire (NorAQ) was used to assess exposure to interpersonal violence in childhood and adulthood (25,26).” In addition to that, we added the answer alternatives for each situation presented in the NorAQ to the methods section: “There were four answer alternatives which were same for all questions: 1) no; 2) as a child (before the age of 18); 3) yes, as an adult (when being 18 years old or older) 4) yes, as a child and as an adult (before and after the age of 18)”. 

6. Page 15, 2nd paragraph – The authors should delete their description of significant findings for age in table 4 since they are borderline.

The authors appreciate this comment. The age variable in Table 4 shows the change in the adjusted odds ratio per one year, which is relatively short period of time. In case we had chosen to use age groups in multinomial regression analyses, the adjusted odds ratios would have been bigger. However, we agree that in previous manuscript the description of this variable could have caused confusion and to make it clearer that AOR is presented for one year we changed the description of the variable.

7. Table 4 – The authors should include the numbers along with the percentages of men and women who were affected by violence. The authors should include a footnote indicating the variables used for adjustment.

Authors are grateful for this comment. Footnote indicating the variables used for adjustment and the numbers of men and women affected by one form of interpersonal violence and PVA were added to the Table 4.

Reviewer III

Major Issues

Introduction:

1. In this paper, the authors make an implicit assumption that family/intimate partner violence and community violence can be aggregated together. A discussion of why this is feasible is missing.

Authors highly appreciate this comment. However, we want to point out that the main objective of this paper was to give a more comprehensive picture about the overall lifetime exposure to interpersonal violence, not to evaluate exposure to violence in different situations (intimate partner versus community violence). Much of the research has focused on only one kind of victimization (Simmons, Wijma, & Swahnberg, 2015), however, this can cause the underestimation of the full burden of violence (Turner, Finkelhor, & Ormrod, 2010).

To get a better overall understanding about the total burden of interpersonal violence, the authors have decided to aggregate family/intimate partner violence and community violence. The authors agree that in the previous manuscript it was poorly explained and we have carried out changes to the introductions as follows: “Most of the research on interpersonal violence has focused on one of the types of interpersonal violence, like violence perpetrated by current of previous partner. However, many researchers have drawn attention to the fact, that leaving out other types of violence, where the perpetrator was somebody else than the intimate partner or that took place in earlier stages of life, results in lower prevalence estimates and a limited understanding about the total burden of violence (Finkelhor, Ormrod, & Turner, 2007b, 2007a; Simmons et al., 2015; Turner et al., 2010). “

Nevertheless, the authors also note the limitation of the used NorVold Abuse Questionnaire, which does not ask questions concerning the perpetrator. To address this, further discussion was added to the limitations of the paper. The authors agree that the patterns and consequences of violence, according to the perpetrator can be very different. 

2. Relatedly, it is unclear why the authors chose to distinguish the three different forms of violence – physical, sexual, and emotional (yet chose to aggregate interpersonal violence).

We agree with this comment and we have added a more detailed discussion explaining these decisions.

The decision to keep different forms of violence separately is associated with the definition of polyvictimization, which was also added to the manuscript: “Polyvictimization refers to the experience of multiple victimizations of different kinds”(Turner, Shattuck, Finkelhor, & Hamby, 2017). The prevalence of emotional, physical and sexual violence was presented to give a better overview to the readers about the formation of the polyvictimization in adulthood variable. In addition to that, this kind of data showing the prevalence of different forms of interpersonal violence across gender on population basis has been completely lacking in Estonia. 

Since in the NorVold Abuse Questionnaire there are no questions regarding the perpetrator of the lifetime violence, we were not able to present this data. This is addressed as one of the limitations of the paper.

3. A discussion of the hypothesized prevalence of violence (e.g. based on other studies in the same region, or IPV studies in Estonia) is missing.

Authors appreciate this comment. We have added data from previous research carried out in Estonia.

“A previous study demonstrated that in Estonia 17.2% of women had been exposed to physical and 4.1% to sexual intimate partner violence during the last year and it was an important contributor to sexual risk behaviour and adverse sexual health outcomes among women of reproductive age in Estonia (Laanpere, Ringmets, Part, & Karro, 2013). A study carried out by European Agency for Fundamental rights showed that since the age of fifteen every third woman had been exposed to physical or sexual violence by partner/non-partner (European Union Agency for Fundamental Rights, 2014).”

Methods:

4. It is unclear why the aim is modified to “to analyze the association between PVA and variables identified in the literature to be associated with a higher risk for exposure to IPV among women” when the introduction sets the paper up to avoid focusing solely on IPV among women. As such, the “independent variables” sub-section needs further substantiation on why these variables were chosen.

The authors agree with this comment. We have carried out changes to the paragraph, where we explain why these independent variables were chosen.

“The second aim was to analyze the association between exposure to one form of violence, PVA and socio-demographic variables identified in the literature to be associated with a higher risk for exposure to interpersonal violence among men and women (Nyhberg 2013, Romans 2007, Balsam 2005, Dovran 2019).”

5. The authors chose to use a multinomial regression model to illustrate correlates of one or more forms of interpersonal violence. However, it is unclear whether an ordinal logistic regression model was explored or feasible. Additionally, there may be different correlates of each form of violence. I suggest modeling each form of violence separately, as a sensitivity analysis, to determine whether this aggregation is feasible. 

We appreciate this comment. In this paper, we have decided to focus on polyvictimization in adulthood and not each form of interpersonal violence separately. This decision was made based on previous literature showing that polyvictimization independently has the strongest effect predicting poor health outcomes. The main focus of this paper was to analyze the differences in the prevalence and patterns of interpersonal violence by gender.

Results:

6. Knowing that older age is associated with violence in adulthood may not be informative (I see the authors recognize this, as noted in the discussion section). That said, there should be a discussion on why lifetime exposure to violence was chosen as the outcome in the multinomial regression models, and not exposure to violence in the past year, in the methods section.

The authors appreciate this comment and considered also using the exposure to violence during last year in the multinomial regression models. However, exposure to violence during last year gives information only about a very limited time period. When assessing exposure to violence during last year or the most recent violent event, it can cause leaving out all events which took place earlier in life. The authors wanted to show in this paper more comprehensive picture about the total burden of violence during lifetime. Authors have added information to the introduction regarding why we decided to use lifetime exposure to interpersonal violence. We also want to point out, that the age structure of men and women included in the analysis was the same. Moreover, we used age as one important control variable in the model.

Discussion:

7. To state that the correlates and patterns of interpersonal violence between men and women are different does not appear to add additional insights to the violence literature. What makes this study different from others conducted in Estonia or other countries in the region, for example?

Authors are very grateful for pointing this out. This is the first study in Estonia and Eastern-European region analyzing the patterns of violence among men and women on a population basis. We have added further discussion regarding the differences we found between men and women in their lifetime exposure to violence. 

In addition to that, research focusing on interpersonal violence polyvictimization in adulthood is very limited. This paper contributes to the violence literature by analyzing the factors associated with exposure to polyvictimization in adulthood. Further discussion was added to the manuscript. 

8. The discussion on the implications of both low response rates and excessive missingness is weak. For instance, it would be helpful to know how the authors would expect the results to change if there was no missingness.

Authors appreciate this comment. However, we want to point out, that the questionnaires of both surveys were quite long, including more than one hundred questions, which could have affected the response rates. The questions concerning violence, which were in the end of the questionnaire, were filled in rather well (over 95% of the respondents answered to the questions concerning exposure to violence), considering the sensitivity of these questions. 

To address the possible implications of the response rates, we added further discussion as follows: 

“The response rates of both surveys, 47% in EWH and 44.5% in EMS were in the range expected and similar response-rate estimations were used when calculating the sample size. In addition to that, the final samples of both EWH and EMS were representative and had no selectivity bias. Previous population-based studies focusing on violence have yielded similar response rates (20,33). The prevalence of interpersonal violence in EWH survey is comparable with the results of the previous studies carried out in Estonia (20,47). However, there is a possibility that the most highly victimized men and women were not able or chose not to participate in the surveys. Based on that, it is possible, that the prevalence of PVA could be even higher than shown in this paper.”

9. As the authors note, in the introduction, interpersonal violence encompasses both family/intimate partner violence and community violence. Since interpersonal violence is broad and types of violence differ between women and men, it may be useful to disaggregate the prevalence of IPV and community violence by gender and discuss these differences, so as to get a better picture of polyvictimization. However, the authors also note perpetrator was not assessed. It would be helpful to discuss this as a limitation of the work.

Authors agree with comment and have added further discussion concerning the limitations associated with not knowing the perpetrator of violence

“Information concerning the perpetrator of the violence could offer better explanations regarding the differences of exposure to different forms of violence among men and women. From previous research it is known than women are more often exposed to systematic forms of intimate partner violence, which could explain higher levels of polyvictimization among them (Johnson, 2011).”

Minor Issues

Introduction:

10. What kind of violence is the authors referring to when they say, “A higher prevalence of violence is associated with a greater acceptance of violence in society…”?

Authors are grateful for pointing out this and we have carried out changes as follows: 

“A higher prevalence of interpersonal violence is related to greater acceptance of violence in society, lower gender equality, and many other societal factors.”

Methods:

11. Are there any differences in responses for electronic vs paper questionnaires?

There were no differences, authors have added this information also to the manuscript. “There were no significant differences between the two responding methods regarding responses about exposure to violence or socio-demographic characteristics.”

Results:

12. The distinction between “unknown” and “missing” in Table 2 (labeled as Table 1) is unclear. I recommend that these two categories be combined.

Authors have considered combining these two, but as the “unknown” shows respondents who have answered but did not know the education of their mother/father we have decided to keep them separated. This can be interpreted as an indicator of the relationship between child and parent when they are not aware of the educational level on their parent.

13. Does the variable “Biological children” refer to having one or more children? If so, I think the latter label is clearer.

We appreciate this recommendation and have made changes as suggested.

14. Did the authors omit “missing” and “unknown” categories from the multinomial regression models? It is unclear whether or not this is the case.

Authors appreciate this comment. Missing values were omitted from the analysis. This information was added to the methods section of the manuscript.

15. Which ages are “younger” and which ages are “older”? This characterization of the variable should be consistent in Table 2 and Table 4.

We appreciate for pointing this out. In the multinomial logistic regression in Table 4 age was used as a discrete variable, in Table 2 age groups were presented. This was not clearly explained in the previous version of the manuscript and authors have added additional information concerning that and changed the label in Table 4.

Discussion:

16. Note that there are no contextual variables included in the study, yet the authors state, “the contribution of this study is identifying contextual and gender specific factor…”

Authors agree with this comment and have replaced “contextual” with “socio-demographic” factors.

General:

17. There are several instances of grammatical errors (e.g. misspellings) and awkward phrasing (e.g. “men had less often higher education”).

Authors have thoroughly checked the spelling and phrasing of the manuscript and carried out corrections.

---

## [Decision Letter · Decision Letter 1]

28 May 2020

PONE-D-19-24414R1

The prevalence and patterns of exposure to interpersonal violence among men and women in Estonia

PLOS ONE

Dear Dr. Lippus,

Thank you for submitting your manuscript to PLOS ONE. After careful consideration, we feel that it has merit but does not fully meet PLOS ONE’s publication criteria as it currently stands. Therefore, we invite you to submit a revised version of the manuscript that addresses the points raised during the review process.

Reviewer 2 recommended to accept your manuscript for publication. However, reviewer 4, raised some minor points. 

We look forward to receiving your revised manuscript.

Kind regards,

Astrid M. Kamperman

Academic Editor

PLOS ONE

Reviewers' comments:

Reviewer's Responses to Questions

**Comments to the Author**

1. If the authors have adequately addressed your comments raised in a previous round of review and you feel that this manuscript is now acceptable for publication, you may indicate that here to bypass the “Comments to the Author” section, enter your conflict of interest statement in the “Confidential to Editor” section, and submit your "Accept" recommendation.

Reviewer #2: All comments have been addressed

Reviewer #4: (No Response)

2. Is the manuscript technically sound, and do the data support the conclusions?

Reviewer #2: (No Response)

Reviewer #4: Yes

3. Has the statistical analysis been performed appropriately and rigorously? 

Reviewer #2: (No Response)

Reviewer #4: Yes

4. Have the authors made all data underlying the findings in their manuscript fully available?

Reviewer #2: (No Response)

Reviewer #4: Yes

5. Is the manuscript presented in an intelligible fashion and written in standard English?

Reviewer #2: (No Response)

Reviewer #4: Yes

6. Review Comments to the Author

Reviewer #2: (No Response)

Reviewer #4: This manuscript uses data from two cross-sectional surveys (both conducted in 2014), one including 1,244 women and the other including 1,590 men, to study gender differences in 1) the prevalence of three types of interpersonal violence in both childhood and adulthood, and 2) how sociodemographic characteristics at the time of the survey and experience of interpersonal violence in childhood are associated with the probabilities of experiencing one or multiple types of violence in adulthood.

This is a generally well-structural manuscript addressing an important topic in a relatively understudied social context (e.g., former Soviet Union countries). The authors have responded to the earlier reviews; I bring up some new points to be addressed in this round to further strengthen the paper.

1. Throughout the paper, I am quite confused about the sample size. Based on page 8, I think the final analytical sample size is 1,244 for women and 1,590 for men. But in Table 4, sample size for men was 1,239 and for women is 1,532. Please check the sample size carefully and be consistent across tables. I also suggest authors including final sample size in the abstract.

2. Page 4: “Exposure to four of more adverse childhood experiences…”; You mean “four or more”?

3. Page 13: The authors have highlighted “Two thirds (n=829, 66.6%) of men and over half (n=862, 54.2%) of women had been exposed to some form of interpersonal violence in their lifetime” as one of major findings several times in the manuscript. But they are not shown in Table 3. I suggest authors adding another column in Table 3 presenting the prevalence of EVER experiencing different forms of violence (either in childhood or adulthood).

4. Maybe I miss it, but percentages of experiencing only one form of violence and PVA are not consistent between Table 3 and Table 4. For example, men who experienced only one form of violence and multiple forms of violence in adulthood is 19.7 (2.8+15.9+1.0) and 2.2 (1.8+0.2+0.2) in Table 3, but it is 28.9% and 5.8% in Table 4.

5. Also in Table 4 which conduct multinomial logistic analyses, the reference group is those who do not report interpersonal violence in adulthood. Since in Table 3, authors have differentiated interpersonal violence in childhood from that in adulthood, it is helpful to include a note in Table 4 clearly indicating who is the “reference group”.

6. In multivariate analysis, exposure to violence in childhood was only measured as a dichotomous variable. Since authors are interested in comparing people who experience one form of violence versus multiple forms in adulthood, it may be also more informative to include categorical variable measuring whether they experience only one violence or multiple violence in childhood as an independent variable.

7. I appreciate authors’ efforts to conduct the first study of interpersonal violence in Estonia, as well in East European countries. In the discussion part, the authors mention that “the results of this study…is in line with previous results from Sweden”(page 16). What about other countries you mentioned in the introduction part, such as US? Are your findings all consistent with other developed countries or is Estonia unique in some ways compared to other contexts studied in the literature? Can the results from Estonia be generalized to other East European countries or other similar social contexts?

7. PLOS authors have the option to publish the peer review history of their article (what does this mean?). If published, this will include your full peer review and any attached files.

Reviewer #2: No

Reviewer #4: No

---

## [Author Response · Author response to Decision Letter 1]

17 Jul 2020

1. Throughout the paper, I am quite confused about the sample size. Based on page 8, I think the final analytical sample size is 1,244 for women and 1,590 for men. But in Table 4, sample size for men was 1,239 and for women is 1,532. Please check the sample size carefully and be consistent across tables. I also suggest authors including final sample size in the abstract.

Authors appreciate this comment. The final sample for analysis included 1244 men and 1590 women. Only in the regression analysis respondents who did not answer some of the questions were excluded. We added additional information regarding that to the methods section under subheading “Analysis”. 

“All respondents who did not answer some of the questions in this study were excluded from the models, the responses of 1239 men and 1532 women were included to the multinomial logistic regression analysis.”

2. Page 4: “Exposure to four of more adverse childhood experiences…”; You mean “four or more”?

Authors are grateful for pointing out this mistake and have corrected it. 

3. Page 13: The authors have highlighted “Two thirds (n=829, 66.6%) of men and over half (n=862, 54.2%) of women had been exposed to some form of interpersonal violence in their lifetime” as one of major findings several times in the manuscript. But they are not shown in Table 3. I suggest authors adding another column in Table 3 presenting the prevalence of EVER experiencing different forms of violence (either in childhood or adulthood).

Authors appreciate this comment highly. We added a new column “Lifetime exposure” to the Table 3, as was suggested by the reviewer. 

4. Maybe I miss it, but percentages of experiencing only one form of violence and PVA are not consistent between Table 3 and Table 4. For example, men who experienced only one form of violence and multiple forms of violence in adulthood is 19.7 (2.8+15.9+1.0) and 2.2 (1.8+0.2+0.2) in Table 3, but it is 28.9% and 5.8% in Table 4.

In table 4 respondents who had been exposed to any form of violence during adulthood or both adulthood and childhood were included in the new variable used in the regression analysis model. Authors agree that this was poorly explained in the methods section and we added further explanation to the methods sections as follows:

“Both, respondents who had been exposed to violence only in adulthood and those who had been exposed to violence both in childhood and adulthood were included in this variable.”

In addition to that, further explanation was added to the results section as follows: 

“Among men, 28.9% (n=358) of the respondents were exposed to one form of interpersonal violence and 5.8% (n=72) to PVA in adulthood or both in childhood and adulthood, among women 17.9% (n=274) and 12.4% (n=190), accordingly.”

5. Also in Table 4 which conduct multinomial logistic analyses, the reference group is those who do not report interpersonal violence in adulthood. Since in Table 3, authors have differentiated interpersonal violence in childhood from that in adulthood, it is helpful to include a note in Table 4 clearly indicating who is the “reference group”.

Authors appreciate this comment. Information regarding the reference category was added to the Table 4. 

6. In multivariate analysis, exposure to violence in childhood was only measured as a dichotomous variable. Since authors are interested in comparing people who experience one form of violence versus multiple forms in adulthood, it may be also more informative to include categorical variable measuring whether they experience only one violence or multiple violence in childhood as an independent variable.

Authors are very grateful for this suggestion and we have changed the variable “Exposure to violence during childhood” as was suggested by the reviewer and we have carried out changes in Table 4 accordingly. 

In association with that, some interesting results surfaced, and changes were carried in the discussion as follows: 

“Some of the factors associated with PVA among men and women were overlapping, such as exposure to violence in childhood and older age, it should be noted that across gender exposure to violence in childhood had a graded relationship with PVA.”

7. I appreciate authors’ efforts to conduct the first study of interpersonal violence in Estonia, as well in East European countries. In the discussion part, the authors mention that “the results of this study…is in line with previous results from Sweden”(page 16). What about other countries you mentioned in the introduction part, such as US? Are your findings all consistent with other developed countries or is Estonia unique in some ways compared to other contexts studied in the literature? Can the results from Estonia be generalized to other East European countries or other similar social contexts?

Authors are very grateful for this comment.

The results of this study are consistent with previous research carried out in other developed countries, which have showed differences across gender. However, most of these studies have been carried out among children or have focused on IPV. We have changed the discussion as follows: 

“Findings from prior studies carried out in other developed countries, which have looked at gender differences in the context of IPV or violence in childhood, have showed different patterns of exposure to violence across gender.”, “…which corroborates the findings of this study” (p. 17).

The authors of this paper believe that as the recent history of Estonia and therefore many aspects of the current social and cultural norms (such as gender equality) are similar in the Eastern European region, the findings of this paper can be generalized to other countries in the region. We have added this also to the discussion section of the paper. 

“In addition to Estonia, these results are particularly useful in the Eastern European region, where until now research demonstrating the prevalence of interpersonal violence has been scarce.”

---

## [Editor Report · Decision Letter 2]

30 Jul 2020

The prevalence and patterns of exposure to interpersonal violence among men and women in Estonia

PONE-D-19-24414R2

Dear Dr. Lippus,

We’re pleased to inform you that your manuscript has been judged scientifically suitable for publication and will be formally accepted for publication once it meets all outstanding technical requirements.

Kind regards,

Astrid M. Kamperman

Academic Editor

PLOS ONE

---

## [Editor Report · Acceptance letter]

5 Aug 2020

PONE-D-19-24414R2 

The prevalence and patterns of exposure to interpersonal violence among men and women in Estonia 

Dear Dr. Lippus:

I'm pleased to inform you that your manuscript has been deemed suitable for publication in PLOS ONE. Congratulations! Your manuscript is now with our production department. 

Kind regards, 

on behalf of

Dr. Astrid M. Kamperman 

Academic Editor

PLOS ONE